# *Linguatula serrata* (Fröhlich, 1789) in Gray Wolf (*Canis lupus*) from Italy: A Neglected Zoonotic Parasite

**DOI:** 10.3390/pathogens11121523

**Published:** 2022-12-12

**Authors:** Donato Antonio Raele, Antonio Petrella, Pasquale Troiano, Maria Assunta Cafiero

**Affiliations:** Istituto Zooprofilattico Sperimentale della Puglia e della Basilicata, Via Manfredonia, 20-71121 Foggia, Italy

**Keywords:** *Linguatula serrata*, Canid wolf, Italy, zoonosis, linguatulosis

## Abstract

*Linguatula serrata*, Frohlich, 1789, is a cosmopolitan zoonotic worm-like parasite of carnivores and other vertebrates including herbivores and omnivores. The adult form of the parasite typically inhabits the upper respiratory system, nares, and frontal sinuses of dogs, wolves, and cats. Infective eggs may be spread by sneezing, nasal secretions, and stool. The immature stages of the parasite are localized in the visceral organs of intermediated hosts, usually ruminants or rodents, and they are orally transmitted to predators during the ingestion of infested viscera. This paper reports the morphological identification and the molecular characterization of *L. serrata* specimen collected from a gray wolf in the Apulia region (southern Italy) and it also provides epidemiological information on this rarely reported zoonosis.

## 1. Introduction

Pentastomids are annulated endoparasites capable of infesting four classes of vertebrates as adults, and both vertebrates and invertebrates as intermediate hosts. Four pentastomid species are of zoonotic interest and, among these, the cosmopolitan *Linguatula* (*L.*) *serrata* (Fröhlich, 1789), also called tongue worm, belongs to the Linguatulidae, the only pentastomid family that uses mammals as definitive hosts [1]. The life cycle of the species, known since 1860, is complex: the adult forms have been usually reported in the nasopharynx of wild and domestic carnivores; instead, ruminants, leporids, swine, and rodents represent intermediate hosts, which may be infested through the contact with pasture or water contaminated by parasite eggs [2]. Humans can serve as accidental intermediate hosts when they ingest eggs (visceral linguatulosis), from which parasitic larvae exit and migrate on viscera. Sometimes, they may act as aberrant final hosts by eating active nymphs in infested poorly cooked animal entrails, leading to the development of a nasopharyngeal form known as *Halzoun/Marrara* syndrome in African, central Asiatic, and eastern Mediterranean countries [2,3,4,5]. In fact, a relatively high presence of immature forms of the parasite was reported from endemic areas in the viscera of slaughtered animals, mainly in the mesenteric lymph, liver, lung, and spleen of domestic ruminants and camels [3,6]. However, very few cases of human infection by adult forms of *L. serrata* have been reported, mainly because the infective larvae usually do not complete their cycle in the human nasopharynx [1,7]. Nevertheless, linguatulosis can cause acute and severe symptoms in humans. This report aims to provide new information on the epidemiology of this zoonotic parasitosis through the molecular characterization of the collected specimens, as well as by discussing the role of wild animals in the infection dissemination.

## 2. Materials and Methods

In May 2021, the Istituto Zooprofilattico Sperimentale della Puglia e della Basilicata (IZSPB) received the carcass of a gray wolf (*Canis lupus*) from the local Veterinary Health Services, recovered in Castelluccio della Daunia municipality (Apulia region, Italy), to ascertain the causes of the death. During the necropsy, a tongue worm was observed in the nasal cavities of the animal (Figure 1). After removal, the parasite was rinsed in PBS solution, placed in a Falcon tube, and sent to the laboratory of parasitology of the IZSPB for analysis (Appendix A). In addition, nasal mucus and stool samples were also collected, and parasitological investigations were carried out by flotation using NaCl-saturated aqueous solutions [8]. The parasite identification was performed by morphological features [9] and confirmed by molecular methods. Briefly, the total genomic DNA was extracted from the specimen using the commercial kit GeneJET Genomic DNA Purification Kit (Thermo Scientific, Vilnius, Lithuania). DNA was used as a template in a PCR targeting the mitochondrial cytochrome c oxidase subunit I (*COX1*), according to the protocol used for generic metazoan detection and identification [10], previously recommended to overcome the genetic diversity and allow for differentiation among the species belonging to the genus *Linguatula* [11]. The PCR products were separated by electrophoresis on 2% agarose gel stained with Sybr^®^ Safe (Thermo Scientific, Milan, Italy) and then visualized by a standard UV transilluminator (Bio-Rad, Milan, Italy). The amplicons were purified by the GeneJET PCR Purification Kit (Thermo Scientific) and their nucleotide sequence was determined using the Sanger technique by the PCR products Big Dye Terminator Kit (Thermo Scientific) at the facilities of Eurofins Genomics (Milan, Italy). The final sequence was assembled and submitted to GenBank with the accession number MW947492.1. A comparison with other sequences from GenBank was performed by BLAST (https://blast.ncbi.nlm.nih.gov/Blast.cgi, 15 February 2021). Representative corresponding sequences, whose details are listed in Table 1, were selected and aligned with the one from this study by the ClustalW algorithm [12], and a neighbor-joining phylogenetic tree was inferred according to the Kimura two-parameter method using the MEGA6 software.

## 3. Results

The collected parasite was tongue-shaped, dorsally slightly convex, and ventrally flattened, with the cuticle transversally striated. It measured 63 mm in length and 8 mm in major width and antero-ventrally presented two pairs of hooks, placed on the sides of the oral aperture (Figure 2). Based on the morphological features, the specimen was identified as a female *L. serrata* showing numerous eggs in ovaries. The tested nasal mucus and feces were found positive for pentastomid eggs. They were ovoid and yellowish, measuring about 90 by 70 µm (Figure 3), and a parasitic embryo, armed with two pairs of hooks, was visible within some of them. The molecular analysis successfully amplified the partial portion of the *COX1* gene from the examined specimen. The BLAST analysis showed that the amplicon was 100% identical to the corresponding region of the NCBI sequences of the *L. serrata* mitochondrial partial *COX1* gene from adult forms collected in infested dogs (accession number MZ052082 and KF029447) (Table 1).The intraspecific variation in the *COX1* sequences for *L. serrata* from GenBank showed low values of differences, as identity ranged from 100% with the gene of an adult sample isolated from a dog in Italy, to 98% with the sequence of a sample collected from cattle in Iran. The phylogeny showed two different clades (Figure 4). The first, quite heterogeneous as hosts and geographical locations, grouped most of the sequences, while the second contained the sequence from this study, a sequence from an adult parasite isolated from a dog in Foggia, Italy, and from another adult collected from a dog in Norway.

## 4. Discussion

We report a case of natural infection by an *L. serrata* adult form in a gray wolf in southern Italy during the animal necropsy. It is well known that adult tongue worms usually inhabit the nasopharynx of mammalian carnivores, mainly wild dogs, cats, foxes, and wolves feeding on the blood and fluids of the final hosts; herbivores, rodents [1,2] and, much more rarely, birds (i.e., seagulls) [12] act as intermediate hosts, being infested through the accidental ingestion of pasture containing parasite eggs. The fecal route represents an important means of contamination for sensitive species, including humans [1], as shown in the described case, where numerous eggs of the parasite were observed in the feces. However, Tasan’s (1987) data confirm previous reports that *L. serrata* eggs are mainly expelled by sneezing; in fact, fecal samples can present negative results despite the observed presence of adult forms in tested dogs [13]. This suggests that a diagnosis reached through a stool exam alone may underestimate the presence of this parasitosis; for this reason, the survey of *L. serrata* in definitive hosts should also include rhinoscopy and/or rhinotomy, as observed in our experience and reported. The dispersion by nasal discharge of immediately infective eggs may more easily contaminate food (but also skin, fur, etc.) than those from visible and obvious fecal sources [14,15], so that the contaminated fur may even represent a potential risk for the owners of infected dogs. The infestation is also considered a food-borne disease in the Middle East and Asian countries [5,16,17] because it is associated with the consumption of poorly cooked meat from infested domestic herbivores, mainly camels and cattle, since these animals are an important source of food in Islamic countries [18,19]. Recent studies conducted on domestic sheep have shown that the average positivity rate among tested animals is about 15% and 5% in Iran [20] and Turkey [21], respectively. In those areas, the high rate of parasite finding coincides with more frequent molecular investigations. On the other hand, cases of linguatulosis in dogs are rare in central and northern Europe, and they are often linked to naturally infested dogs imported from endemic areas of the eastern Europe, such as Romania [22,23,24,25,26]. In Italy, the adult form of the parasite was recently observed in a client-owned dog with nasal carcinoma [27] and previously in stray dogs living in suburban areas [28,29], where they likely become infested by eating contaminated rodents and/or domestic ruminants. Recently, a study carried out on 42 legally hunted gray wolf in Serbia and North Macedonia evidenced parasite infestation in the nasal cavity of a gray wolf from the central Balkans [30]. The visceral linguatulosis by immature forms of the parasite was commonly observed in slaughtered cattle in different European countries, such as Great Britain [31], France [32], and Italy, where the parasitosis was often detected in cattle and sheep during the first half of the 20th century [33,34]. Human cases of linguatulosis have also been reported in Europe: encysted larvae were occasionally observed during autopsies or surgery operations in nodules in different body districts such as the omentum, or excised from the lung [35,36]. Conversely, reports of the larval infestation of humans are much rarer. Larvae were found attached to the mucous membrane of a patient’s cheek [37], or mobile in the eye [38]. It is likely that the scarcity of data about the prevalence of *L. serrata* infection is due to the wide and heterogeneous panel of intermediate hosts, and to the difficulty in investigating wild species, where the predator–prey interactions occurring in the wildlife community may contribute to the spread of the infestation. For this reason, it is important to report cases, including reports on wild animals. Molecular analyses could be useful for diagnosing immature forms scarcely visible in the viscera of intermediate hosts, therefore adding information to achieve a better understanding of the epidemiology of the parasitosis. At the time of this study, molecular strategies seem quite undervalued, as suggested by the presence of only 61 sequences of the *COX1* gene of *L. serrata* in the NCBI database, of which only two are linked to European cases [39]. Among the molecular methods, the analysis of the *COX1* gene remains crucial for detection, identification, and characterization purposes, as well as considering its suitability for phylogeny, as reported by Ghorashi et al. [40]. The study of nucleotide sequences and their submission in shared online databases is crucial to achieve molecular data for further epidemiological studies on the distribution and genetic evolution of this neglected zoonotic parasite.

## 5. Conclusions

The described finding of an adult-form *L. serrata* in a wolf in southern Italy confirms the presence of the parasite two decades after the last described finding in a dog [29] and it suggests the active circulation of the parasite in intermediate hosts, probably domestic and/or wild reservoir animals living in the area. Although the systematic search for visceral forms of linguatulosis would be advisable in all slaughterhouses of domestic ruminants, in free-ranging type farms and in hunted specimens, this approach could be extremely arduous, time consuming, and difficult to be applied. In fact, the encysted nymphs appear as small nodules (a few millimeters in size) and the larvae passages lead to unspecific hemorrhagic pathways in parasitized organs, such as the lungs and liver. However, we assert that the scarcity of information on this zoonotic parasitosis should lead to more attention being paid to this issue. Molecular approaches could be very useful in conducting large-scale investigations. In particular, the molecular survey of immature forms of *L. serrata* could reveal new reservoirs of this parasite, especially in wild fauna, and it could be used to update epidemiological features about the prevalence of linguatulosis.

## Figures and Tables

**Figure 1 pathogens-11-01523-f001:**
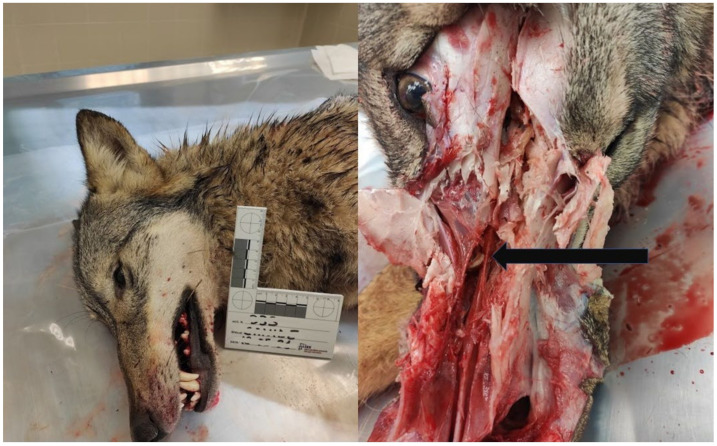
Post mortem examination of a gray wolf: a worm-like parasite (black arrow) firmly attached to the right nasal mucosa of the animal.

**Figure 2 pathogens-11-01523-f002:**
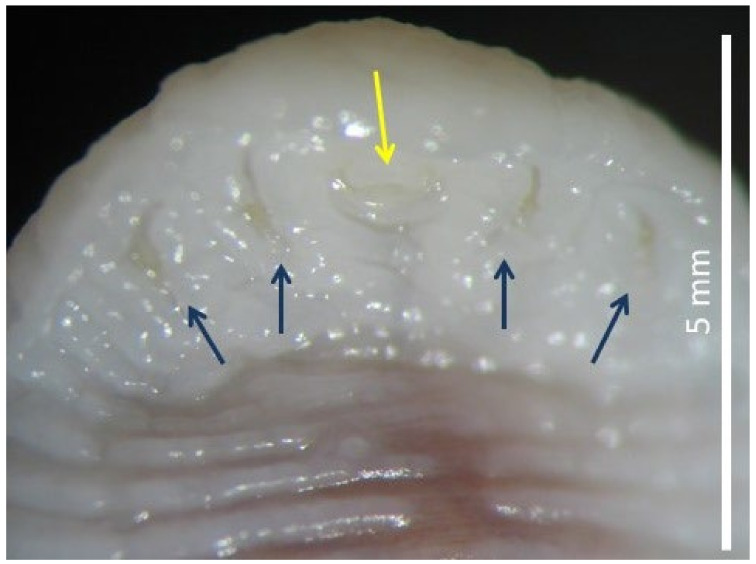
Ventral view of anterior end of *Linguatula serrata* with mouth (yellow arrow) and four hooks (blue arrows).

**Figure 3 pathogens-11-01523-f003:**
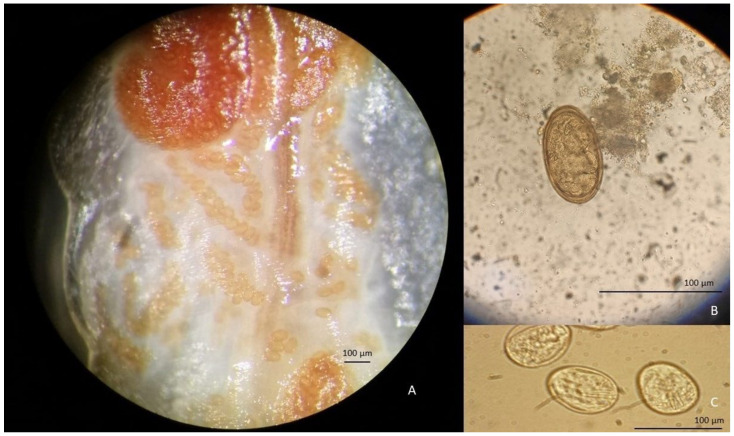
*Linguatula serrata* eggs contained in the ovaries of the collected specimen (**A**) in the stools (**B**) and nasal mucus (**C**) of the gray wolf.

**Figure 4 pathogens-11-01523-f004:**
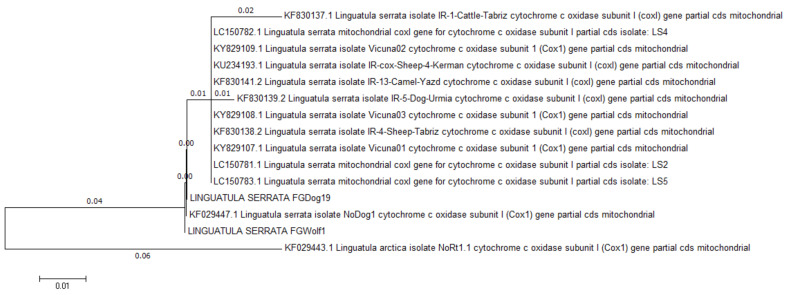
Phylogenetic relationship of *L. serrata* isolates based on partial sequence of *COX1* gene. Distance is shown at branches.

**Table 1 pathogens-11-01523-t001:** Pairwise identity between the partial nucleotide sequence of the *COX1* gene of the isolate from this study (*), and other corresponding sequences from *L. serrata* isolates available in GenBank. *L. arctica* (#) was used as an outgroup.

GenBank Accession Number/Species	Country	Host	% Homology
MW947492/*L. serrata* *	Italy	Wolf	100%
MZ052082/*L. serrata*	Italy	Dog	100%
LC150783/*L. serrata*	Bangladesh	Cattle	99.83%
KF830141/*L. serrata*	Iran	Camel	99.50%
LC150782/*L. serrata*	Bangladesh	Cattle	99.83%
KY829107/*L. serrata*	Peru	Camel	99.67%
KY829108/*L. serrata*	Peru	Camel	99.67%
KY829109/*L. serrata*	Peru	Camel	99.67%
KF830139/*L. serrata*	Iran	Dog	99.00%
LC150781/*L. serrata*	Bangladesh	Cattle	99.83%
KU234193/*L. serrata*	Iran	Sheep	99.52%
KF029447/*L. serrata*	Norway	Dog	100%
KF830138/*L. serrata*	Iran	Sheep	99.50%
KF830137/*L. serrata*	Iran	Cattle	98.00%
KF029443/*L. arctica* #	Norway	Reindeer	89.95%

## Data Availability

Not applicable.

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
