# Peer review of "Linguatula serrata (Fröhlich, 1789) in Gray Wolf (Canis lupus) from Italy: A Neglected Zoonotic Parasite"

_pathogens, 2022, doi:10.3390/pathogens11121523_

Round 1
Reviewer 1 Report
MS pathogens-2073850
Linguatula serrata (Pentastomida: Linguatulidae) in grey wolf
(Canis lupus) from Italy: a neglected zoonosis
by Raele et al.
GENERAL COMMENTS
The ms is a nice contribution aimed to describe a rare case of linguatulosis in a dead wolf.
The topic if of interest; however, there are some aspects which should be better explained/structured throughout the sections.
The ms. is worth publishing after the authors have addressed the following comments and suggestions:
SPECIFIC COMMENTS:
Major comments
Introduction
Line 21: Please add the “(Frölich, 1789) after Linguatula (L.) serrata.
Lines 23-24: “The life cycle of this ….. blood and fluids”. This sentence is unclear. “Currently”? Please, rephrase and report the correct life cycle with definitive (domestic and wild carnivores) and intermediate hosts (ruminants, leporids, swine, rodents). Rodents are mentioned soon after with only herbivores.
Line 30: I would suggest to mention which kind of viscera are involved (liver, spleen, lungs).
Line 31: Linguatulosis in humans have been registered also in Central Asiatic countries (Iran for instance)
Lines 31: The reference 3 seems not completely fitting. The referee would suggest to add references on published cases of linguatulosis in humans. In this view, refer directly to Reference number 10.
Line 33: camels are also ‘domestic ruminants’.
Line 36-38: Authors should report the specific aim: i.e. to present a case of linguatulosis in a wolf; in addition, the present contribution provides “information on the diffusion and epidemiological aspects of this zoonotic parasitosis”.
Line 37: please, select the best verb. ‘Provide’ would be good.
Material and Methods
Line 40: I would suggest to include the Country (Italy).
Line 46: Add that the specimens “was” removed by forceps before being “rinsed”
Line 77-78: I suggest to include how the specimen’s observation was done and how the eggs were retrieved.
Discussion
This section needs to be completely re-organized and harmonized since the contribution deals with a case of linguatulosis in a wolf. I would suggest to start with the specific case, the feature of this finding, including the molecular aspects, explaining the reasons why the wolf could have been affected (including the reason of the wolf death), the diffusion in other canids, also in Italy, leaving tips on epidemiological aspects (including diffusion in intermediate hosts) and the possible zoonotic risk.
Line 106-108: This section is a repetition if it is included into the Introduction too.
Line 108 and 113: I apologize I do not see the relationship of the text with references 12 and 13. Please check and clarify.
Line 110: replace “this way” with ‘fecal route’.
Conclusion
Line 150: “finding [26]”. Add “in a dog”.
Lines 151-153: I find quite difficult such systematic “searching”, considering that in the liver and lungs, only hemorrhagic pathways are found crossing the entire organ; encysted nymphs are detectable as nodules a few millimeters in diameter in the intestinal wall, mesenteric lymph nodes, liver, spleen, and lungs. Are we confident that this practice is possible? Another option, but inapplicable in slaughterhouses, is the serology. The authors should think about and if I am wrong, they can clarify this aspect.
Figure 1
I suggest to remove “a dead” wolf, since the Figure 1’s incipit is: “Post mortem examination”.
Figure 3
I suggest to include the egg size in Fig. 3b. Then, remove “using the flotation test” from the caption.
Table 1
The * and # in the first and final line, respectively, should be explained in a legend.
Other comments
Line 64: remove the dot before the parenthesis in (Table 1)
Line 77-78 and Line 90: Please write “Linguatula serrata” “L. serrata” in italics
Line 19: I suggest to be cautious with phylum Arthropoda, being the classification of this parasite still unclear.
English
Although I am not an English mother tongue, I think the ms. requires revision by a native speaker to eliminate some mistakes or inaccuracies, e.g., lacking of the article in some sentences, improper usage of the third person for some verbs, States, instead of Countries, etc.
Author Response
Reviewer: 1
Comments to the Author
GENERAL COMMENTS
The ms is a nice contribution aimed to describe a rare case of linguatulosis in a dead wolf. The topic if of interest; however, there are some aspects which should be better explained/structured throughout the sections. The ms. is worth publishing after the authors have addressed the following comments and suggestions:
SPECIFIC COMMENTS:
Major comments
Introduction
Line 21: Please add the “(Frölich, 1789) after Linguatula (L.) serrata.
R: Linguatula (L.) serrata (Frölich, 1789) has been added in the paper and the title was also rewritten.
Lines 23-24: “The life cycle of this ….. blood and fluids”. This sentence is unclear. “Currently”? Please, rephrase and report the correct life cycle with definitive (domestic and wild carnivores) and intermediate hosts (ruminants, leporids, swine, rodents). Rodents are mentioned soon after with only herbivores.
R: The life circle of the parasite has been better explained and rewritten. Intermediate hosts have been listed.
Line 30: I would suggest to mention which kind of viscera are involved (liver, spleen, lungs).
R: As suggested, viscera involved during the parasitosis has been inserted in ms.
Line 31: Linguatulosis in humans have been registered also in Central Asiatic countries (Iran for instance)
R: Thanks, a new reference has been added.
Lines 31: The reference 3 seems not completely fitting. The referee would suggest to add references on published cases of linguatulosis in humans. In this view, refer directly to Reference number 10.
R: Thanks for you report, the bibliography has been partially modified and improved.
Line 33: camels are also ‘domestic ruminants’.
R: Although camels also belong to the order Artiodactyla Owen, 1848 the authors accept as true that suborder Tylopoda cannot be included among the infraorder Sheep Flower, 1883.
Line 36-38: Authors should report the specific aim: i.e. to present a case of linguatulosis in a wolf; in addition, the present contribution provides “information on the diffusion and epidemiological aspects of this zoonotic parasitosis”.
R: Thanks, the specific aim has been added.
Line 37: please, select the best verb. ‘Provide’ would be good.
R: The sentence has been rewritten.
Material and Methods
Line 40: I would suggest to include the Country (Italy).
R: Italy has been added.
Line 46: Add that the specimens “was” removed by forceps before being “rinsed”
R: The sentence has been rewritten.
Line 77-78: I suggest to include how the specimen’s observation was done and how the eggs were retrieved.
R: The sentence has been rewritten.
Discussion
This section needs to be completely re-organized and harmonized since the contribution deals with a case of linguatulosis in a wolf. I would suggest to start with the specific case, the feature of this finding, including the molecular aspects, explaining the reasons why the wolf could have been affected (including the reason of the wolf death), the diffusion in other canids, also in Italy, leaving tips on epidemiological aspects (including diffusion in intermediate hosts) and the possible zoonotic risk.
R: We thank the Reviewer for having brought to our attention such an incorrectness. The paragraph, as well as the whole manuscript, has been revised and information has been fixed.
Line 106-108: This section is a repetition if it is included into the Introduction too.
R: The section has been removed.
Line 108 and 113: I apologize I do not see the relationship of the text with references 12 and 13. Please check and clarify.
R: Thanks for you report, the bibliography has been partially modified and improved.
Line 110: replace “this way” with ‘fecal route’.
R: The sentence has been rewritten.
Conclusion
Line 150: “finding [26]”. Add “in a dog”.
R: “in a dog” has been added.
Lines 151-153: I find quite difficult such systematic “searching”, considering that in the liver and lungs, only hemorrhagic pathways are found crossing the entire organ; encysted nymphs are detectable as nodules a few millimeters in diameter in the intestinal wall, mesenteric lymph nodes, liver, spleen, and lungs. Are we confident that this practice is possible? Another option, but inapplicable in slaughterhouses, is the serology. The authors should think about and if I am wrong, they can clarify this aspect.
R: The section was reviewed and re-organized. The sentences: “The search for visceral forms of linguatulosis should therefore be recommended in all slaughterhouses of domestic ruminants, in free-ranging type farms, and in hunted specimens. In fact, the encysted nymphs are detectable as nodules very small (few millimeters) in size and the larvae passages only lead to unspecific haemorrhagic pathways in parasitized organs, such as lungs and liver.” has been added. Certainly, a serological screening in susceptible hosts could be useful as proposed by Jomes and Riley 1991,103: 331-337 as well as molecular surveys.
Figure 1
I suggest to remove “a dead” wolf, since the Figure 1’s incipit is: “Post mortem examination”.
R: the term “a dead” wolf has been removed.
Figure 3
I suggest to include the egg size in Fig. 3b. Then, remove “using the flotation test” from the caption.
R: The size of eggs has been added and “using the flotation test” has been removed from the caption.
Table 1
The * and # in the first and final line, respectively, should be explained in a legend.
R: The legend of the table has been rewritten.
Other comments
Line 64: remove the dot before the parenthesis in (Table 1)
R: The typo has been fixed.
Line 77-78 and Line 90: Please write “Linguatula serrata” “L. serrata” in italics
R: The writing of L. serrata has been corrected.
Line 19: I suggest to be cautious with phylum Arthropoda, being the classification of this parasite still unclear.
R: Although the systematics of Pentastomids is constantly evolving and L. serrata is currently considered an aberrant arthropod, the authors decided to remove the word “arthropod”.
English
Although I am not an English mother tongue, I think the ms. requires revision by a native speaker to eliminate some mistakes or inaccuracies, e.g., lacking of the article in some sentences, improper usage of the third person for some verbs, States, instead of Countries, etc.
R: The manuscript (including abstract) has been carefully and thoroughly revised by a colleague with strong English writing skills. Attention has also been paid to make sentence clearer and more concise.
Reviewer 2 Report
The report of Linguatula serrata the grey wolf in italy provides valuable data. The methods are apropiate, including useful molecular information.The photographs will be helpful for other researchers that have similiar findings.
Minor corrections: line 77-78 and 84 L. serrata should be write it in italics L. serrata
In table 1 Perù should be write it as Peru
Author Response
Reviewer 2
The report of Linguatula serrata the grey wolf in italy provides valuable data. The methods are apropiate, including useful molecular information.The photographs will be helpful for other researchers that have similiar findings.
Minor corrections: line 77-78 and 84 L. serrata should be write it in italics L. serrata
In table 1 Perù should be write it as Peru
- The misstatements have been fixed.
Reviewer 3 Report
This manuscript provides scientific information and useful results for an epidemiological update.The manuscript is well written too, evertheless, there are some flaws to be assessed before it can be published.
I am not English mother tongue and I have the feel that the English should be revised.
I envision that Authors would be able to amend the ms, in order to make it suitable for publication in Pathogenes after minor revision.
Minor revision
Material and methods
Line 48 - Authors should report reference in flotation test and which flotating solution was used
Line 53 - Is the molecular protocol used reported in Shamsi et al.? To specify the reference after the gene
Line 61 - Authors should report in the results that the sequences have been deposited in GeneBank
Results
Lines 77, 84, 87 - Authors should write L. serrata in cursive font
Line 82 - “…from each specimen”, how many specimens were examined? “the sequences were identical to each other” , what does it mean? DNA was extracted by how many specimens? Rewrite, please.
Author Response
Reviewer 3
This manuscript provides scientific information and useful results for an epidemiological update.The manuscript is well written too, evertheless, there are some flaws to be assessed before it can be published. I am not English mother tongue and I have the feel that the English should be revised.I envision that Authors would be able to amend the ms, in order to make it suitable for publication in Pathogenes after minor revision.
R: The manuscript (including abstract) has been carefully and thoroughly revised by a colleague with strong English writing skills. Attention has also been paid to make sentence clearer and more concise.
Minor revision
Material and methods
Line 48 - Authors should report reference in flotation test and which flotating solution was used.
R: Particulars of the flotation technique have been included in the manuscript
Line 53 - Is the molecular protocol used reported in Shamsi et al.? To specify the reference after the gene
R: The cited reference has been added.
Line 61 - Authors should report in the results that the sequences have been deposited in GeneBank
R: Details of the obtained sequence have been added.
Results
Lines 77, 84, 87 - Authors should write L. serrata in cursive font
R: typos has been fixed.
Line 82 - “…from each specimen”, how many specimens were examined? “the sequences were identical to each other” , what does it mean? DNA was extracted by how many specimens? Rewrite, please.
R: The sentence has been rewritten.
Reviewer 4 Report
.

Author Response
Reviewer 4
GENERAL COMMENTS
I revised the case report entitled “Linguatula serrata (Pentastomida: Linguatulidae) in grey wolf (Canis lupus) from Italy: a neglected zoonosis” submitted for publication in the Journal “Pathogens” [Manuscript ID: pathogens-2073850]. In this manuscript, the authors report the morphological and molecular identification of a Linguatula serrata specimen collected in a grey wolf in the Apulia region, Southern Italy, and provide epidemiological information on linguatulosis. Overall, I found this report interesting and well‐written. It provides new insights on linguatulosis in Southern Italy after the last paper was published more than 20 years ago. The manuscript is well-constructed and clear. The introduction and the discussion cover the subject exhaustively and the video, in my opinion, increases the quality of the work by showing clear evidence of what the authors have found and identified. The methods are convincing and reliable, although additional information regarding the microscopic identification of the specimen should be provided (see specific comment). The use of the English language is generally good; however, some misspellings and inaccuracies have been spotted throughout the text (see some examples in the specific comments) and a further revision carried out by a native English speaker is highly recommended. The manuscript would be of great interest for the readers of “Pathogens”. Therefore, the manuscript could be published after the minor issues listed below are addressed and revisions of the text are made.
R: We would like to thank the reviewer for the suggestion, the manuscript has been revised by a colleague with strong English writing skills. Attention has also been paid to make sentence clearer and more concise.
SPECIFIC COMMENTS
- Line 11: There seems to be an extra “s” between are and orally.
R: The sentence has been rewritten.
- Lines 32, 111: “In fact” is two words.
R: The typos has been fixed.
- Lines 48/49: How was the morphological identification performed? Were specific identification
keys used? They should be cited.
R: The reference has been added.
- Line 49: “in basis to” is incorrect.
R: The sentence has been rewritten.
- Lines 14, 55: Provides and allows, respectively.
R: The sentence has been rewritten.
- Lines 81/82: Up until this point of the manuscript it would seem that the specimen found and isolated
was only one. However, the fact that this sentence states “…from each specimen.” makes the reader
wonder whether there was more than one. Please clarify.
R: The sentence has been rewritten and the discrepancy has been resolved.
- Lines 84, 86, 87, 90: Linguatula serrata should be written in italics.
R: The typos has been fixed.
- Line 96: Comma instead of semicolon after letter (A).
R: the semicolon has been removed.
- Line 105: “result be” is wrong.
R: The sentence has been rewritten.
- Line 114: I suggest rephrasing this sentence since Islam is a religion, not a geographical area.
R: Sorry for the inaccuracy, obviously the phrase has been rewritten.
- Lines 118-120 This paragraph about European Countries and imported cases is confusing because
Romania belongs to Europe so the sentence “…dogs imported into Europe from endemic areas such
as observed in Romania” is unclear and, for the same principle, so is the next (lines 120-122).
R: Sentences have been rewritten.
- Lines 120, 125: Replace contrary with on the contrary.
R: Sentences have been rewritten.
- Line 154: Replace revel with reveal.
R: the verb has been removed and the sentence rewritten
- The reference list and the citations should be double checked as it seems there is something wrong:
the last reference of the list (Ghorashi et al., 2016) is not numbered and comes after number 37
(Gjerde, 2013). However, in the text (line 143), the authors state “…as also reported by Ghorashi et
- [37]” meaning that Ghorashi is reference 37 and not 38 like it would seem. It is worth double
checking.
R: During the final preparation of the document the line assignment system has divided the reference. The error has been corrected and the bibliography has been updated.